# Assessment of Immunogenic and Antigenic Properties of Recombinant Nucleocapsid Proteins of Five SARS-CoV-2 Variants in a Mouse Model

**DOI:** 10.3390/v15010230

**Published:** 2023-01-13

**Authors:** Alexandra Rak, Nikolay Gorbunov, Valeria Kostevich, Alexey Sokolov, Polina Prokopenko, Larisa Rudenko, Irina Isakova-Sivak

**Affiliations:** 1Department of Virology, Institute of Experimental Medicine, Saint Petersburg 197022, Russia; 2Department of Molecular Genetics, Institute of Experimental Medicine, Saint Petersburg 197022, Russia

**Keywords:** SARS-CoV-2, nucleocapsid phosphoprotein, monoclonal antibody, immunogenicity, recombinant protein, cross-reactivity, epitopes

## Abstract

COVID-19 cases caused by new variants of highly mutable SARS-CoV-2 continue to be identified worldwide. Effective control of the spread of new variants can be achieved through targeting of conserved viral epitopes. In this regard, the SARS-CoV-2 nucleocapsid (N) protein, which is much more conserved than the evolutionarily influenced spike protein (S), is a suitable antigen. The recombinant N protein can be considered not only as a screening antigen but also as a basis for the development of next-generation COVID-19 vaccines, but little is known about induction of antibodies against the N protein via different SARS-CoV-2 variants. In addition, it is important to understand how antibodies produced against the antigen of one variant can react with the N proteins of other variants. Here, we used recombinant N proteins from five SARS-CoV-2 strains to investigate their immunogenicity and antigenicity in a mouse model and to obtain and characterize a panel of hybridoma-derived monoclonal anti-N antibodies. We also analyzed the variable epitopes of the N protein that are potentially involved in differential recognition of antiviral antibodies. These results will further deepen our knowledge of the cross-reactivity of the humoral immune response in COVID-19.

## 1. Introduction

During almost three years of the global pandemic caused by SARS-CoV-2, the virus genome has evolved significantly, generating multiple lineages and variants that have spread readily around the world [1,2,3]. Moreover, such variability in the viral antigenic properties has resulted in a significant decrease in COVID-19-vaccine effectiveness against genetically evolved SARS-CoV-2 variants and caused widespread vaccine breakthrough infections [4,5]. This is mainly because the vast majority of COVID-19 vaccines elicit Spike-specific antibodies, but the Spike viral protein is highly variable and easily mutates to escape population immunity [6,7]. In contrast to the Spike protein, the viral nucleocapsid (N) protein is highly conserved among all SARS-CoV-2 variants and has 90% similarity to the SARS-CoV-1 N protein [8]. High levels of N-specific IgG antibodies are induced upon COVID-19, since N is the most abundantly expressed SARS-CoV-2 protein [9]. Furthermore, the N protein is one of the major targets for virus-specific T-cell responses [10], which makes this antigen a promising target for COVID-19 vaccine development [11]. Indeed, it was shown that a Spike-based vaccine supplemented with the N protein conferred acute protection in both the lung and the brain after a challenge, while a Spike-based vaccine alone provided acute protection only in the lung [12].

N is a flexible and multivalent RNA-binding protein that contains three dynamic disordered regions: the N-terminal domain (NTD), the linker and the C-terminal domain (CTD). These regions undergo liquid–liquid phase separation when mixed with RNA [9]. A central β-hairpin of the N molecule contains a serine/arginine-rich region (residues 176–209) that serves as a regulatory element [13]. A recent study found that the N protein can be readily identified on the surfaces of SARS-CoV-2-infected and surrounding cells, where it is bound via electrostatic high-affinity binding to heparan sulfate and heparin [14]. This surface localization of the N protein makes it accessible to anti-N antibodies, which activate Fc-receptor-expressing innate immune cells. Furthermore, N is able to bind, with high affinity, to 11 human chemokines, including CXCL12β: a leukocyte chemotaxis factor, inhibited by N, from SARS-CoV-2, SARS-CoV-1 and MERS-CoV [14].

The N protein is an important antigen for development of COVID-19 diagnostics. First of all, this antigen can be directly detected in biological fluids for the purpose of diagnosis in the early stages of infection. The specificity and relatively high sensitivity of this direct analysis has been shown in fluorescence immunochromatographic (FIC) assays [15] and ELISA tests [16,17]. Moreover, strong positive correlation was observed between elevated plasma N-antigens and odds of pulmonary damage severity, resulting in worsened clinical outcomes [17,18]; therefore, N-level measurement upon hospital admission may improve risk stratification through identification of patients with implicit odd of severe diseases [18]. An N-antigen-based assay may be performed in a simple self-test format, although its sensitivity and diagnostic accuracy would be lower compared to those of an RT-PCR assay [19]. On the other hand, the N protein may be used as a basis for ELISAs that detect antiviral antibodies and are similar in specificity and sensitivity to those based on S-protein fragments. Full-length and truncated forms are suitable for development of test systems; the latter showed greater sensitivity in analysis of mouse, rabbit and human sera and seems to be a better serological marker for evaluating SARS-CoV-2 immunogenicity [20]. It was shown that antibodies against S and N proteins in COVID-19 convalescents were both detectable for up to 200 days after a positive SARS-CoV-2 RT-PCR test [21], but demonstrated markedly different trends in signal intensity: anti-N antibodies were characterized with lower persistence [22].

Despite the highly conservative nature of the N protein, it still undergoes slow evolutionary changes, which can potentially affect its tertiary structure [23]. As a consequence, the sensitivity and specificity of N-based COVID-19 diagnostic tools may be compromised. Previously, we observed cross-reactivity of anti-N antibodies, raised to the ancestral SARS-CoV-2 virus, through demonstration of a strong positive correlation in the magnitudes of anti-N (B.1) antibodies and antibodies specific to various variants of concern (VOCs) [24]. However, little is known about the immunogenicity and antigenicity of the N proteins of these VOCs or whether slight antigenic differences can affect the performances of N-based diagnostic tools. Here, we assessed the immunogenicities of the recombinant N proteins of five SARS-CoV-2 strains belonging to different lineages, as well as cross-reactivity of induced anti-N antibodies. In addition, we generated and characterized several monoclonal antibodies (mAbs) raised to the N protein of the B.1 virus, with different epitope specificities. Our results revealed the varied recognition repertoire of antiviral antibodies generated in response to immunization with the N proteins of different VOCs and cross-reactivity of anti-N (B.1) mAbs.

## 2. Materials and Methods

### 2.1. Cells, Viruses and Proteins

African green monkey kidney Vero E6 cells were obtained from the American Type Culture Collection (ATCC) and maintained in DMEM supplemented with 10% fetal bovine serum (FBS) and 1× antibiotic–antimycotic (AA) (all from Capricorn Scientific, Ebsdorfergrund, Germany). Three SARS-CoV-2 viruses were obtained from the Smorodintsev Research Institute of Influenza (Saint Petersburg, Russia): HCoV-19/Russia/StPetersburg-3524/2020 (B.1 Lineage, Wuhan)**,** hCoV-19/Russia/SPE-RII-32759S/2021 (B.1.617.2 Lineage, Delta) and hCoV-19/Russia/SPE-RII-6243V1/2021 (B.1.1.529 Lineage, Omicron). These viruses were grown on Vero E6 cells using DMEM supplemented with 2% FBS, 10 mM of HEPES and 1× AA (all from Capricorn Scientific, Ebsdorfergrund, Germany) at 37 °C and 5% CO_2_. After full cytopathic effect was reached, the virus-containing media was harvested, clarified via low-speed centrifugation and stored at −70 °C in aliquots. All experiments with live SARS-CoV-2 were performed in a biosafety-level-3 laboratory.

Recombinant N proteins were expressed in *Escherichia coli* cells as previously described [24], using full-length sequences of the following SARS-CoV-2 strains:hCoV-19/Russia/StPetersburg-3524/2020 (B.1 Lineage, Wuhan);hCoV-19/Russia/SPE-RII-27029S/2021 (B.1.351 Lineage, Beta);hCoV-19/Japan/TY7-503/2021 (P.1 Lineage, Gamma);hCoV-19/Russia/SPE-RII-32759S/2021 (B.1.617.2 Lineage, Delta);hCoV-19/Russia/SPE-RII-6243V1/2021 (B.1.1.529 Lineage, Omicron).

### 2.2. Mice

All experiments were performed in compliance with relevant laws and institutional guidelines and approved by the local Ethical Committee of Institute of Experimental Medicine (protocol No.1/22, dated 18 February 2022). Female CBA and BALB/c mice that weighed 18 to 20 g were purchased from the Stolbovaya breeding nursery (Moscow region, Russia). The CBA mice were immunized intraperitoneally three times with 20 μg of the recombinant N protein in an AlumVax adjuvant (1:1 *v*/*v*) (OZ Biosciences, San Diego, CA, USA) at 14-day intervals. Blood samples were collected 14 days after the final immunization, and sera were stored at −20 °C.

### 2.3. Assessment of Virus-Specific Antibodies in Mouse Serum Samples

Serum IgG antibodies specific to N proteins were measured with an ELISA. Briefly, high-binding 96-well plates (Thermo Fisher Scientific, Waltham, MA, USA) were coated with purified recombinant N proteins, 100 ng per well, in a carbonate–bicarbonate buffer (pH 7.4) overnight at 4 °C. Then, the plates were blocked with 1% BSA in PBS (pH 7.4) for 40 min at 37 °C and washed 3 times with PBS-T (PBS with 0.1% Tween 20). Serum samples were diluted 5-fold in PBS-T (1:500 to 1:121,500) and added to wells, followed by incubation for 1 h at 37 °C. Each sample was tested in duplicate. After washing, an HRP-conjugated goat antimouse IgG secondary antibody (Bio-Rad, Hercules, CA, USA) was added to each well and incubated at 37 °C for 1 h. Then, the plates were finally washed and developed with 1-Step TMB Substrate Solution (HEMA, Moscow, Russia) for 15 min. After the reaction with 1 M of H_2_SO_4_ was stopped, the resulting absorbance was measured at a wavelength of 450 nm (OD_450_) using an xMark Microplate Spectrophotometer (Bio-Rad, Hercules, CA, USA). The area under the OD_450_ curve (AUC) values were calculated as a trapezoidal square for each serum sample and expressed in arbitrary units.

### 2.4. Monoclonal Antibodies

Production of mAbs was obtained through methods of hybridoma technology [25]. BALB/c mice were immunized with 10 μg of the N protein (B.1), emulsified with a complete Freund’s adjuvant, in plantar aponeurosis of the hind limbs. After 4 weeks, the animals were immunized subcutaneously with 10 μg of the N protein (B.1) mixed with an incomplete adjuvant. On the 30th day after the second immunization, the animals were boosted intravenously with 5 μg of the N protein (B.1) in saline. The lymphocytes of the inguinal and abdominal lymph nodes were isolated 4 days later and mixed with Sp 2/0 myeloma cells at a ratio of 2:1. Cell fusion was performed in prewarmed 50% 1500 kDa polyethylene glycol (PEG) for 1.5 min, followed by dropwise addition of an equal volume of RPMI-1640 medium. After hybridization, cells were pelleted and cocultured with peritoneal macrophages (at a 10:1 ratio) in 96-well culture plates. To select hybridomas, we used a RPMI-1640 medium that contained 10% FBS, 10^−4^ M of hypoxanthine, 4 × 10^−7^ M of aminopterin and 1.6 × 10^−5^ M of thymidine. Primary screening of clones was performed via ELISAs of culture media from hybridomas that used the recombinant N protein (B.1) as a coating antigen. Then, the ability of the mAbs to recognize the N proteins of different SARS-CoV-2 strains was evaluated using the same ELISAs. Hybridomas that produced specific antibodies against N proteins were then subcloned, and individual clones were expanded in 175 cm^2^ flasks for intraperitoneal injection into animals (2 million cells per mouse). Isotype determination was performed using an ISO-2 kit (Sigma, St. Louis, MO, USA) according to the instructions of the manufacturer. Antibodies were purified from the ascitic fluid of mice, which was collected 12–18 days after intraperitoneal injection of hybridoma cells through protein-A-affinity chromatography with the MabSelect sorbent (GE HealthCare, Chicago, IL, USA), using the manufacturer’s protocol.

### 2.5. SDS-PAGE and Western blot

Sodium dodecyl sulfate–polyacrylamide gel electrophoresis (SDS-PAGE) was used to check the structure and purity of obtained anti-N mAbs, while the ability of anti-N mAbs to detect linear epitopes of N proteins of five SARS-CoV-2 strains was assessed via western blotting. Purified mAbs and recombinant N proteins were resolved in reduced conditions on a 10% polyacrylamide gel at 120 V for 1 h before being stained with colloidal Coomassie G-250 solution (Bio-Rad, Hercules, USA) for 1 h at room temperature or semi-dry transferred to 0.45 μm nitrocellulose membranes for 2 h at 100 V. Blots were blocked overnight at 4 °C with 5% skimmed milk in PBS-T and then treated with anti-N mAbs diluted 1:1000 in block buffer for 1 h at 37 °C. Then, goat anti-mouse HRP-conjugated secondary antibody (Bio-Rad, Hercules, CA, USA) diluted 1:3000 in blocking solution was added to the triple-washed blots for 1 h at 37 °C. After three washes with PBS-T, the blots were developed with 0.05% solution of diaminobenzidine (Sigma, St. Louis, MO, USA) in PBS containing 1% hydrogen peroxide. Finally, the membranes were washed with water and the images were captured using Gel Doc EZ Gel Documentation System (Bio-Rad, Hercules, CA, USA).

### 2.6. Cell ELISA and Immunocytochemistry

A cell-based ELISA was used to check the specificities of purified mAbs to native viral antigens. A 14C2 monoclonal antibody (Abcam, Cambridge, UK) that binds the M2e protein of the influenza virus was used as a negative control. For the assay, 96-well plates were seeded with 4 × 10^4^ Vero E6 cells per well the day before virus inoculation. Cell monolayers were rinsed twice with PBS prior to inoculation with 50 μL of SARS-CoV-2 diluted in DMEM supplemented with 2% FBS, AA and 10 mM of HEPES to reach a multiplicity of infection (MOI) of 0.01. After adsorption at 37 °C for 1 h, the cells were overlaid with 100 µL of the same culture medium and incubated at 37 °C for 1 or 3 days for B.1 or P.1/B.1.1.529 viruses, respectively. After incubation, the culture medium was carefully removed and the cells were fixed with 2% formalin in PBS at 4 °C overnight. Then, the fixative solution was removed and the plates were washed with PBS-T and blocked with 3% skim milk in PBS-T for 1 h at 37 °C. Then, 50 μL of 3-fold dilutions of mAbs in PBS-T (from 15 to 0.02 µg/mL) were added to the wells, and the plates were incubated for 1 h at 37 °C. After being washed with PBS-T, the plates were treated with a 1:3000 solution of goat antimouse immunoglobulins conjugated with horseradish peroxidase (Bio-Rad, Hercules, CA, USA) for 1 h at 37 °C. Finally, the plates were thoroughly washed and stained with 1-Step TMB Substrate Solution (HEMA, Moscow, Russia). The resulting OD was measured using a BioRad Model 680 microplate reader (Bio-Rad, Hercules, CA, USA) at a wavelength of 450 nm.

Immunocytochemical analysis was performed in a similar way. Vero E6 cells were seeded on 6-well tissue-culture plates at a dose of 1 × 10^6^ cells per well. After 2 days of incubation with B.1, P.1 or B.1.1.529 viruses, these cells were fixed and sequentially treated with 5 µg/mL of mAbs and goat antimouse HRP-conjugated immunoglobulins. Then the plates were stained with 3-amino-9-ethylcarbazole (Sigma, St. Louis, MO, USA) in the presence of 1% H_2_O_2_, according to the manufacturer’s protocol.

### 2.7. Statistical Analysis

Data were analyzed using the statistical tool of GraphPad Prism 6.0 Software. Compliance with normal distribution was checked with the Shapiro–Wilk test. The Wilcoxon matched-pairs test was used to compare serum IgG antibody responses to different N antigens. One-way ANOVA with the Tukey post hoc test was used to examine the significance of differences between several study groups. The significance level was set at *p* < 0.05.

## 3. Results

### 3.1. Assessment of Immunogenicity of N Proteins

To investigate the immunogenicity and antigenicity of the N proteins of the ancestral SARS-CoV-2 strain and different VOCs, we immunized mice with recombinant N proteins of five SARS-CoV-2 strains [24] and measured the magnitudes of N-specific IgG responses, and the ability of these antibodies to cross-react with other N antigens, using an in-house ELISA protocol. Testing serum samples with homologous N antigens revealed significant differences in the levels of induced anti-N antibodies in different immunization groups; the highest IgG levels were noted in mice immunized with the ancestral B.1 (Wuhan) antigen compared to VOCs B.1.351 (Beta), P.1 (Gamma), B.1.617.2 (Delta) and B.1.1.529 (Omicron) (Figure 1). The strongest differences in immunogenic properties were noted between the N protein of the ancestral B.1 (Wuhan) virus and the N proteins of the P.1, B.1.617.2 and B.1.1.529 VOCs (Figure 1b, *p* < 0.0001). Since all of the N proteins were expressed and administered to mice in an identical manner, the variable levels of the anti-N antibodies were most likely due to the differences in protein sequences and the content of the linear and/or spatial B-cell epitopes located in the N proteins.

To further characterize the antigenic properties of the N proteins of the five SARS-CoV-2 variants, we compared the magnitudes of IgG responses, in mice immunized with each N protein, against all five recombinant N antigens: B.1 (Wuhan), B.1.351 (Beta), P.1 (Gamma), B.1.617.2 (Delta) and B.1.1.529 (Omicron). We used the area under the OD_450_ curve (AUC) values as a measure of intensity of antibody binding to the N antigen, then normalized the AUC values of each heterologous antigen to the AUC data for homologous antigens. This artificial parameter allowed assessment of cross-reactivity of mouse immune sera against various N antigens, regardless of the magnitude of the response. Interestingly, the antibody that was raised to the N (B.1) protein bound to the heterologous N antigens to a lesser extent, suggesting accumulation of escape mutations in this protein with virus evolution (Figure 2A). Similarly, the antibodies induced via the N (B.1.351) protein were less likely to be recognized by the N proteins of SARS-CoV-2 variants that circulated at later times (Figure 2B). In general, the intensity of antibody binding from the heterologous N antigens was reduced compared to that of the homologous protein (Figure 2), although several exceptions were noted: the N (B.1) protein bound to the IgG raised to the N (B.1.351) protein better than did the homologous antigen (Figure 2B); the N (B.1.617.2) antigen recognized a higher proportion of antibodies raised to the N (P.1) protein (Figure 2C); and the N (B.1.351) antigen revealed higher levels of IgG raised to the N (B.1.1.529) protein (Figure 2E). These data indicate that different N proteins can generate different subsets of N-specific antibodies; this should be taken into account as a limitation in development of N-based vaccines and serology assays. In particular, our data from the N (P.1), N (B.1.617.2) and N (B.1.1.529) immunization groups suggest that N-specific antibodies raised to infection with evolutionarily diverse SARS-CoV-2 will be poorly recognized by the N antigen from the ancestral B.1 strain, which is currently present in the majority of N-based serological tests (Figure 2C,D,E).

### 3.2. B- and T-cell N-Protein Epitopes, including Substitutions

To understand which differences in the amino acid sequences of N proteins can be associated with diverse recognition of N-specific antibodies raised to various N antigens, we aligned the N-protein sequences of five SARS-CoV-2 strains, mapped their unique substitutions on the N proteins on the structural scheme (Figure 3) and listed all substitutions in Appendix A. Furthermore, using the Immune Epitope Database (IEDB) resources, we identified linear B-cell (Appendix A) and T-cell (Appendix A) epitopes that contained these mutations.

The epitope coverage of SARS-CoV-2 proteins is well-studied: 15,905 viral epitopes have been deposited in IEDB, including 1159 N-protein epitopes, which have been described in 96 studies. In total, 791 of these N-protein epitopes have been identified as B-cell-mediated immune-response targets. The majority of B-cell epitopes of the N protein identified to date have been deposited at the IEDB based on the studies of Hotop et al. [26], Mishra et al. [27], Heffron et al. [28], Gregory et al. [29] and Schwarz et al. [30]. A total of 174 N-protein epitopes were confirmed in T-cell assays and are class I epitopes described as having protective potential in humans by Tarke et al. [31] and Heide et al. [32] and in transgenic mice by Zhuang et al. [33].

Noteworthily, there was no linear accumulation of mutations in the N protein over time, since each VOC has its own set of mutations compared to the ancestral B.1 (Wuhan) variant (Appendix A), suggesting that all studied SARS-CoV-2 lineages evolved independently. Nevertheless, a number of mutations seem to correlate with the immunogenicity of the recombinant N protein (Figure 1). Thus, the highest numbers of mutations were found in the antigens of the B.1.1.529 and B.1.617.2 variants (Figure 3), which affected a high proportion of the linear B-cell epitopes established for the B.1 N protein (Appendix A). A closer analysis of the cross-reactivity of the N-specific antibodies induced via each recombinant N protein (Figure 2) and the amino acid differences between the particular N immunogens and each N protein used as an antigen, carried out in ELISA (Appendix A), suggested that the residues at positions 13, 80, 203 and 204 had the greatest influence on N-protein antigenicity. Interestingly, the deletion of the three amino acid residues at positions 31–33 did not result in significant impairment of the antigenic properties of the N protein; sera from mice immunized with the P.1 (Beta) and B.1.617.2 (Delta) N proteins recognized the B.1.1.529 (Omicron) N antigen at the same level as the homologous antigens, and vice versa (Figure 2). Notably, the substitutions at positions 13, 203 and 204 were noted for several VOCs, and given that they can significantly change the antigenicity and/or immunogenicity of the N protein, it can be assumed that they are major escape mutations that drive the evolution of SARS-CoV-2.

### 3.3. Generation and Characterization of Anti-N Monoclonal Antibodies

To further characterize the antigenic properties of the N proteins of different VOCs, we generated four mAbs (NCL2, NCL5, NCL7 and NCL10) through immunization of BALB/c mice with the N protein (B.1), using standard hybridoma technology. Purified anti-N antibodies were isotyped as follows: NCL2 and NCL10 were of the IgG1 isotype, while NCL5 and NCL7 were of the IgG2a isotype. We characterized these mAbs using the SDS-PAGE (Figure 4a) and Western blot (Figure 4b) analyses. The electrophoretic mobility of the mAbs corresponded to the expected molecular weights (~160 kDa and ~55 + 25 kDa in nonreducing and reducing conditions, respectively). The SDS-PAGE analysis of the purified mAbs demonstrated the absence of significant amounts of contaminating components (Figure 4a). For study of the specificity of the newly generated mAbs, those mAbs were used in the WB analysis to detect recombinant N proteins; there, an additional recombinant N protein of seasonal coronavirus OC43 was used, along with five SARS-CoV-2 antigens (Figure 4b). Interestingly, NCL2, NCL5 and NCL7 recognized all of the SARS-CoV-2 N antigens but not the OC43 N protein, whereas the NCL10 antibody did not recognize the N protein of the B.1.1.529 strain but was able to bind to the N protein of seasonal coronavirus OC43 (Figure 4b). These data suggest that the NCL10 antibody recognized B-cell epitopes with mutations specific to the Omicron variant only, most probably the region with the deletion of residues 31–33 (Figure 3). In addition, mutation P13L could be responsible for the altered recognition of the NCL10 antibody. However, the N-terminal region of the N protein differs significantly between the OC43 and SARS-CoV-2 viruses; therefore, it remains to be elucidated exactly which epitope is targeted by the NCL10 mAb.

Detection of native N antigens, which are present in virus-infected cells, using the cell ELISA approach recapitulated the results of the Western blot analysis: while NCL2/5/7 mAbs universally recognized the B.1, P.1 and B.1.1.529 virus-infected cells, the NCL10 antibody was unable to bind to the B.1.1.529 SARS-CoV-2-infected cells (Figure 5a). Moreover, the OD_450_ values, which were determined based on the concentration of the resulting virus–antibody complexes, were different between NCL10 and the three remaining mAbs when identical antibody concentration was used, suggesting that the NCL10 antibody binds with lower intensity to N antigens, probably due to the lower affinity of this binding. An immunocytochemical analysis further demonstrated that NCL10 mAb does not recognize the N protein of the B.1.1.529 VOC (Figure 5b). Future studies with generations of escape mutants after passaging SARS-CoV-2 variants in the presence of generated mAbs will allow the precise epitope mapping thereof.

## 4. Discussion

The impact of N-protein mutations on the reliability of data obtained with approved test systems has not yet been sufficiently studied [31]. In addition, development of N-based COVID-19 vaccines also requires studies of immunogenicity and the possibility of generations of anti-N antibodies with a putative autoimmune effect [32]. Previously, we established an in-house ELISA protocol based on the recombinant N proteins of different SARS-CoV-2 VOCs to evaluate the cross-reactivity of N-specific antibodies in COVID-19 convalescents [24]. A strong positive correlation in the magnitude of anti-N (B.1) antibodies and those of antibodies specific to four other VOCs in COVID-19-recovered patients was found, suggesting that N-binding antibodies are highly cross-reactive and the most immunogenic epitopes within this protein are not under selective pressure. Here, we assessed the antigenicities and immunogenicities of N proteins of different VOCs in a mouse model through comparison of the magnitudes and the cross-reactivities of the anti-N antibodies generated in response to immunization with the recombinant N proteins. The strongest impact on the magnitude of homologous antibody responses was revealed for the N proteins of the P.1, B.1.617.2 and B.1.1.529 strains (Figure 1 and Figure 2). The amino acid sequences of these antigens are known to contain a large number of mutations compared to that of the ancestral B.1 strain (Figure 3), and the linear epitopes that carry them have been identified as targets for antiviral antibodies through multiple B-cell assays (Appendix A). Thus, we hypothesize that differences in the specificities of anti-N antibodies produced in response to immunization with similar antigens may be due to evolutionarily determined variability in immunogenic epitopes, leading to emergence of escape mutations. T-cell epitopes of N proteins that contain variable regions may also serve as triggers of T-cell-mediated immune response, according to the results of the activation and binding analyses (Appendix A). Most of these epitopes were shown to bind to recombinant human MHC molecules of the HLA-A*01:01, HLA-A*02:01 and HLA-B*07:02 alleles. Although we did not assess T-cell responses in the mouse model, the presence of mutations within the established T-cell epitopes of various HLA alleles may indicate variability of N-protein-induced, T-cell-mediated immune response in humans infected with different SARS-CoV-2 strains.

During the first two years of the pandemic (2019–2020), the main detectable mutations in the SARS-CoV-2 genome were D614G (81.5%) in the S protein and a combination of R203K/G204R (37%) in the N protein [33]. Later, the adaptive nature of the R203K/G204R mutations in the N proteins of the P.1 and B.1.1.529 variants was shown in silico, and large-scale phylogenetic analysis indicated that the R203K/G204R was associated with the elevated transmissibility and infectivity of the B.1.1.7 SARS-CoV-2 variant. A positive correlation has been found between COVID-19 severity and frequency of 203K/204R [34]. In addition, the R203K/G204R combination appeared to increase nucleocapsid phosphorylation resistance via inhibition of the GSK-3 kinase, resulting in increased virus replication. Similar occasional alanine substitutions also occurred at positions 203 and 204 and led to the same outcomes, indicating an evolutionary trend towards ablation of the ancestral RG motif to increase SARS-CoV-2 infectivity [35]. The virus-like particles that bore the RG203/204KR mutations in the N protein stimulated an augmented humoral immune response and enhanced neutralization in immunized mouse sera [36]. COVID-19 induced with viruses with R203K and G204R resulted in inferior clinical outcomes [37]. The 203/204 mutations occurred in the phosphorylated “RS-motif” [38], which is localized within the serine-rich region of 181–213: a target for phosphorylation that allows recruitment of the host RNA helicase DDX1, thus facilitating template readthrough and synthesis of longer subgenomic mRNA [38].

The results of a docking analysis indicated that the site for binding of targeted antiviral agents is localized in the region of 66–134 in the RNA-binding domain of the N protein [39]. Therefore, mutations in this region, such as P80R in the P.1 SARS-CoV-2 N protein, may influence the sensitivity of the virus to these drugs.

P13L/S substitution unique to the N proteins of the B.1.1529 and B.1.351 strains is located in the CD8 immunodominant region that is targeted by cells that exhibit central and effector memory phenotypes [40]. Furthermore, peptides that contain this substitution have been identified as positive in an HLA-binding analysis of MHC class II epitopes [41] and as targets for antiviral antibodies in COVID-19 convalescents [42]. R203M mutation is localized into peptides for which specific binding of anti-SARS-COV-2 antibodies was detectable [26]. Peptides that contained residue at 205 elicited interferon-γ production in convalescent donors [43], while peptides with residues at 215 and 377 showed specific Ab binding and IFN releasing as a result of T-cell stimulation [27,41,44]. Epitopes that included amino acid residues at 31–33, which are absent in the B.1.1.529 variant, exhibited low-intensity binding of Abs but stimulated IFNγ release in COVID-19 convalescents [26,44]. Peptides that included residue of the N protein at 63 provided strong IFNγ release, as was demonstrated with cell sorting and ELISPOT assays in COVID-19 convalescents and peptide-vaccinated people [45]. P80R mutation is localized in peptides that provide binding, T-cell activation and IFNγ release after in vivo administration or restimulation [44,46,47].

Thus, mutations P13L, ERS31-33 del, D63G, R203K/M, G204R and D377Y, unique to the P.1, B.1.617.2 and B.1.1.529 strains, can be considered variations that appeared in the viral genome as a result of attempts to evade host immune response. Published data on peptides that contain variations that uniquely distinguish the N proteins of different SARS-CoV-2 strains from each other, as well as our results on the varying specificities of anti-N antibodies produced in response to immunization with N proteins of different VOCs, allowed us to consider these substitutions as escape mutations that arose in the viral genome as a result of immune pressure.

It should be noted that our study was limited to assessment of N-based humoral immune responses but not T-cell immunity. As the mouse major histocompatibility complex (MHC) is known to be remarkably different from the human leukocyte antigen (HLA) system [48], our animal model is not applicable for assessment of T-cell-mediated immune responses triggered via N-protein immunization. To study T-cell-involving reactions with maximal approximation to the human organism, use of HLA transgenic humanized mouse models is required [49,50].

To additionally test N-protein antigenicity and obtain a useful tool for detection of viral antigens, we also generated, in this work, a number of mAbs against the N protein of the B.1 strain and examined their specificities towards different VOCs. Previously, several mAbs against the N proteins of different VOCs have been obtained and described. Hodge et al. compared the properties of flexible and rigid anti-N antibodies and showed the utility of the latter for the highly sensitive ELISA detection of the N protein NTD [51]. In another study, a panel of 41 mAbs against the N protein of the B.1 strain was obtained in order to develop latex-based lateral flow immunoassays (LFIAs). These test systems allowed detection of as low as 8 pg of a purified protein or 625 TCID_50_/mL of a virus, and they cross-reacted with the P.1 and B.1.617.2 variants [52]. The use of N-specific detection by highly sensitive nanobodies C2 and E2, which are specific to B.1, P.1 and B.1.617.2, was demonstrated by Isaacs et al. [53]. Molecular modeling of the interaction of anti-N mAbs specific to 501Y.V1-V3, obtained by Yamaoka et al. [54], revealed binding with exterior protein surface epitopes, and immunochromatographic test systems, combined with silver amplification technology, were developed. Lee et al. produced mAbs against conserved N-protein peptides and developed a laboratory-confirmed sandwich ELISA as a rapid biosensor, allowing detection of as low as 4 × 10^3^ TCID_50_/reaction for SARS-CoV-2 or 1 ng/mL of the recombinant N protein [55]. An original immunochromatographic test-strip method based on the 14 anti-N mAbs was proposed as a tool for detection of the B.1 and B.1.617.2 variants [56].

Our obtained mAbs, named NCL2, NCL5 and NCL7, appear to be suitable for detection of N proteins in all five SARS-CoV-2 variants, whereas the NCL10 mAb was unable to recognize the N protein of the B.1.1.529 strain (Figure 4b and Figure 5). Notably, the latter mAb was also able to recognize the recombinant N protein of seasonal coronavirus OC43; although this property does not allow use of NCL10 for diagnostic purposes, it indicates the presence of similar epitopes in the molecules of the two evolutionarily diverse viruses, and this mAb may be applicable for studying the structures and evolution of SARS-CoV-2 proteins. Our further studies will be devoted to development of test systems based on the obtained mAbs and defining the affinity of their interaction with viral antigens.

Overall, our results demonstrate the slow-evolving nature of the SARS-CoV-2 N protein, which affects the specificity profile of anti-N antibodies and should be considered as a limitation in development of N-based vaccines and test systems.

## Figures and Tables

**Figure 1 viruses-15-00230-f001:**
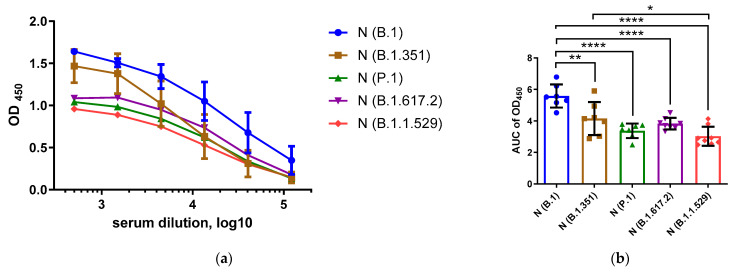
Assessment of homologous anti-N antibody levels in serum samples of mice immunized with B.1 (blue), B.1.351(brown), P.1 (green), B.1.617.2 (magenta) and B.1.1.529 (red). Hereafter, the color designations of the N proteins of different SARS-CoV-2 VOCs are the same. The OD_450_ values are shown in (**a**), and the area under the OD_450_ curve (AUC) values are shown in (**b**). These data were compared using ANOVA with the Tukey post hoc test. * *p* < 0.05, ** *p* < 0.01, **** *p* < 0.0001.

**Figure 2 viruses-15-00230-f002:**
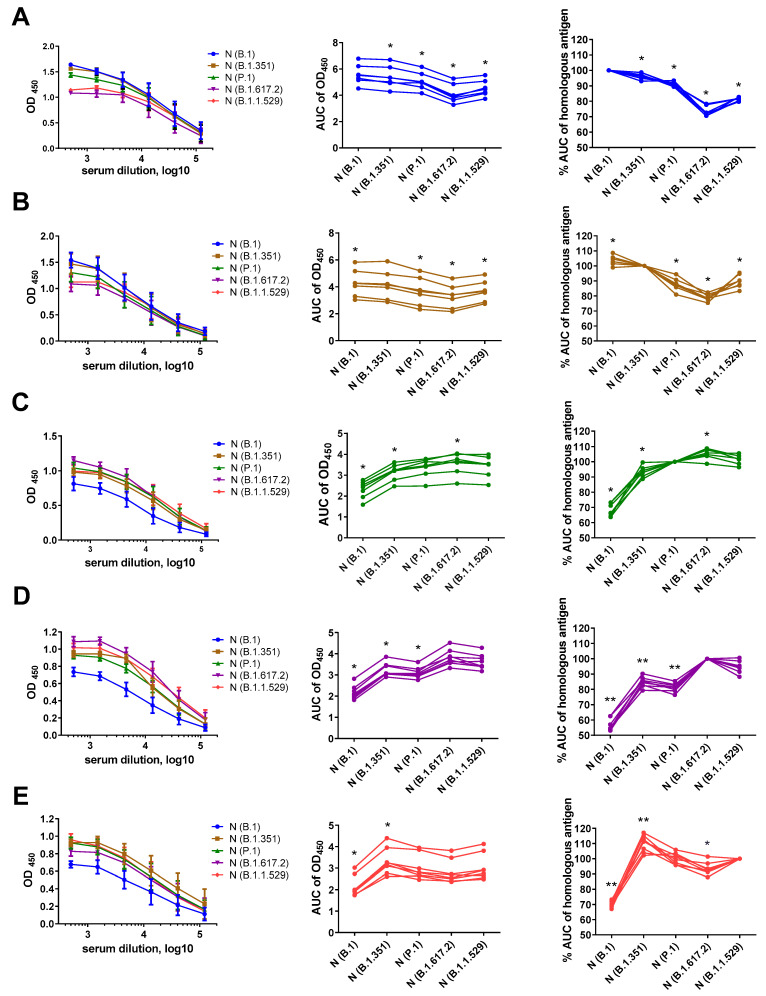
N-based ELISAs of serum samples of mice immunized with B.1 (**A**, blue), B.1.351 (**B**, brown), P.1 (**C**, green), B.1.617.2 (**D**, magenta) and B.1.1.529 (**E**, red). The left panel shows the mean OD_450_ values in ELISAs with the indicated N antigen. The area under the OD_450_ curve (AUC) values for each serum sample tested against five N antigens are shown in the middle panel. The right panel shows the AUC values normalized to the homologous antigen. These data were compared using the Wilcoxon matched-pairs test. * *p* < 0.05, ** *p* < 0.01.

**Figure 3 viruses-15-00230-f003:**
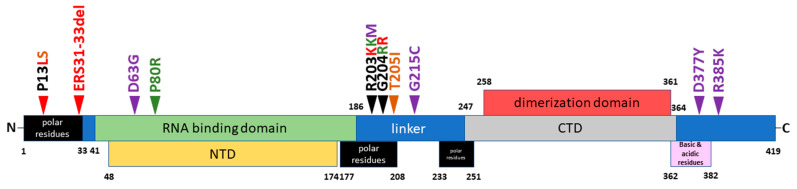
Scheme of N-protein domains and variable amino acid residues characterizing five different variants of the SARS-CoV-2 virus used in this work. The locations of the functional domains of the N protein are given according to [9]. The substitutions present in the N-protein sequences of the B.1.351 (brown), P.1 (green), B.1.617.2 (magenta) and B.1.1.529 (red) strains are indicated with arrows.

**Figure 4 viruses-15-00230-f004:**
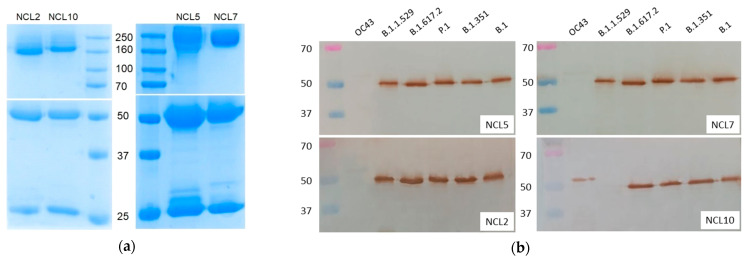
SDS-PAGE (**a**) and Western blot (**b**) analyses of the purified anti-N antibodies. The upper panel of the SDS-PAGE shows the results of the analysis in nonreducing conditions, and the lower panel shows the results of the analysis in the presence of 1% β-mercaptoethanol.

**Figure 5 viruses-15-00230-f005:**
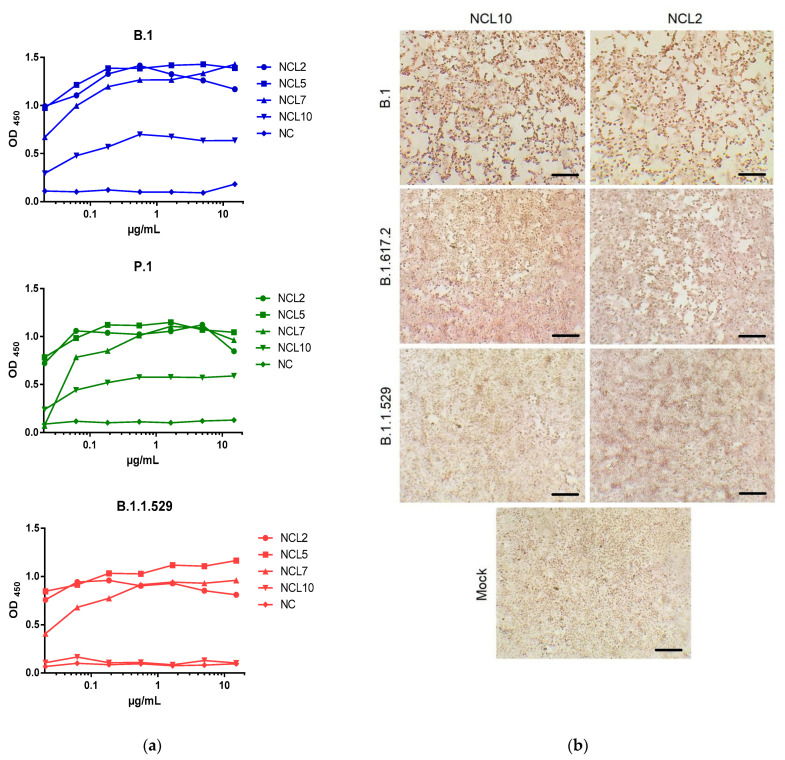
Analysis of specificity of N-specific mAbs in a Vero E6 cell model. *(***a**) Dependence of absorbance intensity (OD_450_) on the concentration (µg/mL) of anti-N mAbs or 14C2 mAbs (NC, anti-influenza M2e antibodies). Vero E6 cells were infected with the B.1, B.1.617.2 or B.1.1.529 SARS-CoV-2 variant, followed by cell ELISA with indicated mAbs. (**b**) Immunocytochemical analysis of SARS-CoV-2 N protein in infected Vero E6 cells after 48 h of incubation. The different patterns of cytopathic effects caused by three different strains of SARS-CoV-2 are visualized. The positive red staining was developed as a result of treatment with the NCL2 and NCL10 (except for the B.1.1.529 detection) mAbs. The mock comprised the untreated cells. Scale bars: 100 µm.

## Data Availability

The data presented in this study are available on request from the corresponding authors.

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
