# Peer review of "Assessment of Immunogenic and Antigenic Properties of Recombinant Nucleocapsid Proteins of Five SARS-CoV-2 Variants in a Mouse Model"

_viruses, 2023, doi:10.3390/v15010230_

Round 1

Reviewer 1 Report

The authors in this work have accessed the antigenicity and immunogenicity of the SARS-CoV2 protein N, a promising target for the development of serology tests and vaccines against COVID-19. Given its conservative nature, when compared to the current commonly used protein spike, the protein N has interesting perspectives of study, as it would potentially determine a lower number of vaccine-escaping mutants. Among other results, they have seen that differences in protein N coming from different variants would produce divergent sets of antibodies against the virus. The reason for that is observed in slightly differences observed in the protein N of the different VOCs. The work is really interesting and is up to be published upon some minor changes that I suggest below: 

  • Line 131: Change to “The production of mABs was obtained...” 

  • Line 157: “seeded with 4x10e5 Vero E6 cells...” : are you sure? That is a lot of cells for a well in a 96-well plate.  

  • Figure 5a: I would recommend the authors continue with the pattern of colors for each VOC used in previous figures. 

  • Why did you have not done any experiments concerning the T-cell responses given you already had the mice? That would be very interesting to see in your work.

Author Response

The authors in this work have accessed the antigenicity and immunogenicity of the SARS-CoV2 protein N, a promising target for the development of serology tests and vaccines against COVID-19. Given its conservative nature, when compared to the current commonly used protein spike, the protein N has interesting perspectives of study, as it would potentially determine a lower number of vaccine-escaping mutants. Among other results, they have seen that differences in protein N coming from different variants would produce divergent sets of antibodies against the virus. The reason for that is observed in slightly differences observed in the protein N of the different VOCs. The work is really interesting and is up to be published upon some minor changes that I suggest below:

Authors’ response: we thank the reviewer for the positive feedback and are appreciate all the suggested changes.

  • Line 131: Change to “The production of mABs was..”

Authors’ response: the indicated sentence has been modified.

  • Line 157: “seeded with 4x10e5 Vero E6 cells...” : are you sure? That is a lot of cells for a well in a 96-well plate.

Authors’ response: we are grateful to the reviewer for noting this typo. Indeed, the seeding dose was 4x10e4 per well, which is approximately 4x10e6 per plate.

  • Figure 5a: I would recommend the authors continue with the pattern of colors for each VOC used in previous figures.

Authors’ response: we agree with this suggestion and have modified Figure 5a accordingly.

  • Why did you have not done any experiments concerning the T-cell responses given you already had the mice? That would be very interesting to see in your work.

Authors’ response: we thank the reviewer for reising this important point; however, the scope of this study was limited to the assessment of antibody immune responses, because studying of cross-reactivity of N-based T-cell responses will require adequate animal models, such as humanized or HLA-transgenic mice, or some well-established human PBMC in vitro models. As we used CBA mice in this study, no relevant conclusions would have been drawn from the assessment of mouse-specific T-cell responses. We added the following statement to the Discussion section to address this issue: “It should be noted that our study was limited to the assessment of N-based humoral immune responses, but not T-cell immunity. As mouse major histocompatibility complex (MHC) is known to be remarkably different from human leukocyte antigen (HLA) system [48], our animal model is not applicable for the assessment of T cell-mediated immune responses triggered by N protein immunization. To study T cell involving reactions with maximal approximation to the human organism, the use of HLA transgenic humanized mouse models is required [49,50].”.

Reviewer 2 Report

Article: Assessment of immunogenic and antigenic properties of recombinant nucleocapsid proteins of five SARS-CoV-2 variants in a mouse model. This is an interesting and potentially important manuscript about  recombinant N proteins from five SARS-CoV-2 strains to investigate their 17 immunogenicity and antigenicity in a mouse model.  However, I would suggest some minor modifications.

Page 3.

L. 101: " E. coli cells" to "Escherichia coli cells" 

Page 9.

L. 324: " b – immunocytochemical analysis of SARS- 324 CoV-2 N protein in infected Vero E6 cells after 48 h of incubation. Scale bars: 100 μm."  Describe the differences between the images.

L. 327-333: "SARS-CoV-2 N protein is a promising target for COVID-19 serology testing and vaccine development because of its much more conservative nature compared to the commonly used Spike protein antigen or its RBD domain [11]. Nevertheless, the N protein is also prone to mutations, and their impact on the reliability of data obtained with approved test systems has not yet been sufficiently studied [34]. In addition, the development of N- based COVID-19 vaccines also requires studies on immunogenicity and possibility of generation anti-N antibodies with a putative autoimmune effect [35]." This is Introduction or literature review. 

Author Response

Article: Assessment of immunogenic and antigenic properties of recombinant nucleocapsid proteins of five SARS-CoV-2 variants in a mouse model. This is an interesting and potentially important manuscript about recombinant N proteins from five SARS-CoV-2 strains to investigate their 17 immunogenicity and antigenicity in a mouse model. However, I would suggest some minor modifications.

Authors’ response: we thank the reviewer for the positive feedback and are appreciate all the suggested changes.

Page 3.

  1. 101: "E. coli cells" to "Escherichia coli cells" 

Authors’ response: the "E. coli cells" has been replaced with more precise one.

  1. 324: " b – immunocytochemical analysis of SARS- 324 CoV-2 N protein in infected Vero E6 cells after 48 h of incubation. Scale bars: 100 μm."  Describe the differences between the images.

Authors’ response: the description of the differences between the images has been added to the Figure 5 capture.

  1. 327-333: "SARS-CoV-2 N protein is a promising target for COVID-19 serology testing and vaccine development because of its much more conservative nature compared to the commonly used Spike protein antigen or its RBD domain [11]. Nevertheless, the N protein is also prone to mutations, and their impact on the reliability of data obtained with approved test systems has not yet been sufficiently studied [34]. In addition, the development of N- based COVID-19 vaccines also requires studies on immunogenicity and possibility of generation anti-N antibodies with a putative autoimmune effect [35]." This is Introduction or literature review.

Authors’ response: we thank the reviewer for this critique. The text containing in the paragraph has been modified accordingly.